# Low-Molecular-Weight Fucoidan as Complementary Therapy of Fluoropyrimidine-Based Chemotherapy in Colorectal Cancer

**DOI:** 10.3390/ijms22158041

**Published:** 2021-07-27

**Authors:** Ching-Wen Huang, Yen-Cheng Chen, Tzu-Chieh Yin, Po-Jung Chen, Tsung-Kun Chang, Wei-Chih Su, Cheng-Jen Ma, Ching-Chun Li, Hsiang-Lin Tsai, Jaw-Yuan Wang

**Affiliations:** 1Division of Colorectal Surgery, Department of Surgery, Kaohsiung Medical University Hospital, Kaohsiung Medical University, Kaohsiung 80756, Taiwan; baseball5824@yahoo.com.tw (C.-W.H.); googoogi05@gmail.com (Y.-C.C.); ajaxyin@gmail.com (T.-C.Y.); glaudiotennis@gmail.com (P.-J.C.); tsungkunchang@gmail.com (T.-K.C.); lake0126@yahoo.com.tw (W.-C.S.); kmu880402@gmail.com (C.-J.M.); dobird05@yahoo.com.tw (C.-C.L.); chunpin870132@yahoo.com.tw (H.-L.T.); 2Department of Surgery, Faculty of Medicine, College of Medicine, Kaohsiung Medical University, Kaohsiung 80756, Taiwan; 3Graduate Institute of Clinical Medicine, College of Medicine, Kaohsiung Medical University, Kaohsiung 80756, Taiwan; 4Division of General and Digestive Surgery, Department of Surgery, Kaohsiung Medical University Hospital, Kaohsiung Medical University, Kaohsiung 80756, Taiwan; 5Department of Surgery, Kaohsiung Municipal Tatung Hospital, Kaohsiung Medical University, Kaohsiung 80145, Taiwan; 6Graduate Institute of Medicine, College of Medicine, Kaohsiung Medical University, Kaohsiung 80708, Taiwan; 7Center for Cancer Research, Kaohsiung Medical University, Kaohsiung 80708, Taiwan; 8Center for Liquid Biopsy and Cohort Research, Kaohsiung Medical University, Kaohsiung 80708, Taiwan; 9Master Program for Clinical Pharmacogenomics and Pharmacoproteomics, School of Pharmacy, Taipei Medical University, Taipei City 11031, Taiwan

**Keywords:** low-molecular-weight fucoidan, colorectal cancer, HCT116 cell, Caco-2 cell, fluoropyrimidine-based chemotherapy, complementary therapy

## Abstract

This study investigated the roles of low-molecular-weight fucoidan (LMWF) in enhancing the anti-cancer effects of fluoropyrimidine-based chemotherapy. HCT116 and Caco-2 cells were treated with LMWF and 5-FU. Cell viability, cell cycle, apoptosis, and migration were analyzed in both cell types. Potential mechanisms underlying how LMWF enhances the anti-cancer effects of fluoropyrimidine-based chemotherapy were also explored. The cell viability of HCT116 and Caco-2 cells was significantly reduced after treatment with a LMWF-–5FU combination. In HCT116 cells, LMWF enhanced the suppressive effects of 5-FU on cell viability through the (1) induction of cell cycle arrest in the S phase and (2) late apoptosis mediated by the Jun-N-terminal kinase (JNK) signaling pathway. In Caco-2 cells, LMWF enhanced the suppressive effects of 5-FU on cell viability through both the c-mesenchymal–epithelial transition (MET)/Kirsten rat sarcoma virus (KRAS)/extracellular signal-regulated kinase (ERK) and the c-MET/phosphatidyl-inositol 3-kinases (PI3K)/protein kinase B (AKT) signaling pathways. Moreover, LMWF enhanced the suppressive effects of 5-FU on tumor cell migration through the c-MET/matrix metalloproteinase (MMP)-2 signaling pathway in both HCT116 and Caco-2 cells. Our results demonstrated that LMWF is a potential complementary therapy for enhancing the efficacies of fluoropyrimidine-based chemotherapy in colorectal cancers (CRCs) with the wild-type or mutated *KRAS* gene through different mechanisms. However, in vivo studies and in clinical trials are required in order to validate the results of the present study.

## 1. Introduction

Colorectal cancer (CRC) is reported to be the third most common type of malignancy and the third leading cause of cancer-related death worldwide [1]. In 2017, approximately 1.8 million new diagnoses of CRC and 896,000 CRC-related deaths were estimated globally [2]. In Taiwan, CRC has been the most common type of malignancy and the third leading cause of cancer-related death since 1996. In 2000 and 2017, the incidence was 32.38 and 66.32 per 100,000, respectively (with 7213 and 16,408 new diagnoses, respectively). Moreover, 6436 people died of CRC in 2019. The mortality rates were 27.3 and 20.6 per 100,000 in 2019 and 2009, respectively [3].

Fucoidan is an algal fucose-enriched sulfated polysaccharide molecule which is extracted from the extracellular matrix (ECM) of various types of brown seaweeds [4]. The main backbone of fucoidan is composed of (1→3)-linked α-l-l-fucopyranosyl residues or of alternating (1→3)- and (1→4) α-l-linked-l-fucopyranosyl residues [5,6]. Fucoidan also contains various quantities of other monosaccharaides, such as D-galactose, D-xylose, D-mannose, and uronic acid [5,6]. Fucoidan exhibits many biological activities, including anti-coagulant, antioxidant, anti-inflammatory, anti-proliferative, anti-diabetic, and anti-cancer activities [5,6,7,8,9,10,11]. The biological activities of fucoidan are reported to be associated with its structure and molecular weight [10,12]. Low-molecular-weight fucoidan (LMWF) has been reported to exhibit greater biological activity relative to high-molecular-weight fucoidan (HMWF) [5,10,12]. Hyperglycemia has been reported to be associated with risk and prognosis of CRC [13]. The anti-cancer activities of fucoidan have been revealed for various types of cancers, including hepatocellular carcinoma, lung cancer, breast cancer, CRC, pancreatic cancer, prostate cancer, and melanoma cancer [5,6,7,8,9,10]. 

Fluorouracil (5-FU) has been used as one of the main therapeutic agents for CRC since the 1990s [14]. Fucoidan exerts the anti-cancer activity, and LMWF is more biologically active. In our previous prospective, a randomized and double-blind controlled trial [15], 60 patients with metastatic CRC (mCRC) were enrolled and divided into two groups, namely a study group (n = 30) and control group (n = 30), and the aim was to evaluate the effects of LMWF when used as a supplement to chemotarget agents. Combination therapy of the FOLFIRI regimen (5-FU, folinic acid, and irinotecan) and bevacizumab (5 mg/kg) were administered biweekly as the first-line therapeutic regimen in all patients. Moreover, 4 g of LMWF (extracted from *Sargassum hemiphyllum*) was administered to patients in the study group. The disease control rate was significantly higher in the study group (92.8% vs 69.2%, *p* = 0.026). Therefore, the benefit of supplemental therapy with LMWF was demonstrated; this therapy improved the disease control rate of patients with CRC [15]. Considering the results of our previous study [15], we hypothesized that LMWF enhances the anti-cancer effects of fluoropyrimidine-based chemotherapy. Patients with stage III CRC and positive expression of epidermal growth factor receptor (EGFR) have worse prognoses [16] and may require additional agents to enhance the efficacy of fluoropyrimidine-based chemotherapy. Therefore, we performed an in vitro study to investigate the roles of LMWF in enhancing the anti-cancer effect of fluoropyrimidine-based chemotherapy. Moreover, we attempted to identify the possible mechanisms underlying how LMWF enhances the anti-cancer effect of fluoropyrimidine-based chemotherapy.

## 2. Results

### 2.1. Effect of LMWF and 5-FU on Cell Viability

We observed a significantly lower viability of HCT116 and Caco-2 cells at 24 h (all *p* < 0.05; Figure 1A and Figure 2A, respectively) after treatment with LMWF or 5-FU compared with control cells. Moreover, HCT-116 and Caco-2 cells exhibited significantly lower cell viability at 24 h (all *p* < 0.05; Figure 1A and Figure 2A, respectively) after treatment with the combination of LMWF and 5-FU compared with control cells, cells treated with LMWF, and those treated with 5-FU.

### 2.2. Effects of LMWF and 5-FU on Cell Cycle

In HCT116 cells, the cell cycle distribution did not change significantly after treatment with LMWF (Figure 1B). Moreover, after 5-FU treatment, the proportion of sub-G1 and G0/G1 cells was significantly increased, and the proportion of G2M cells was significantly decreased (*p* < 0.01). However, the proportion of S-phase cells did not change significantly after treatment with 5-FU. After treatment with the combination of LMWF and 5-FU, the proportion of S-phase cells was significantly increased compared with those treated with LMWF or 5-FU (*p* < 0.01). However, in Caco-2 cells (Figure 2B), the cell cycle distribution did not significantly differ after treatment with LMWF, 5-FU, or the combination of LMWF and 5-FU (all *p* > 0.05).

### 2.3. Effects of LMWF and 5-FU on Apoptosis

In HCT116 cells, fucoidan did not significantly induce early and late apoptosis, whereas 5-FU significantly induced late apoptosis (*p* < 0.01). Moreover, LMWF significantly enhanced late apoptosis caused by 5-FU (*p* < 0.01, Figure 1C). In Caco-2 cells, LMWF and 5-FU induced significantly late apoptosis (*p* < 0.05); however, LMWF did not significantly enhance the effects of 5-FU on late apoptosis induction (Figure 2C).

### 2.4. Signaling Pathways Responsible for LMWF and 5-FU Effects on Apoptosis in HCT116 Cells

The mutation statuses of HCT 116 colon cancer cells are KRAS mutation (p.G13D) and phosphatidylinositol-4,5-bisphosphate 3-kinase catalytic subunit alpha (PIK3CA) mutation (p.H1047R) [17]. The Jun-N-terminal kinase (JNK) signaling pathway has been reported to be associated with apoptosis [18,19,20]. Activated JNK can cause the phosphorylation of mitochondrial proteins (Bcl-2 and Bcl-xl) to then induce apoptosis [19]. The inhibition of JNK has been reported to be associated with resistance to chemotherapeutic drugs in various human cancer cells [19]. Moreover, JNKs are associated with the regulation of both intrinsic and extrinsic apoptosis processes [20]. Therefore, in the present study, we investigated whether the JNK signaling pathway is involved in the apoptosis caused by the combined administration of LMWF and 5-FU. The expression of the phosphorylated JNK (p-JNK) protein was significantly increased in cells at 6 h after treatment with the combination of LMWF and 5-FU relative to both cells and cells treated with LMWF or 5-FU (all *p* < 0.05; Figure 3). However, the expression of the total JNK protein did not significantly increase relative to cells treated with LMWF or 5-FU. Cleaved poly (ADP-ribose) polymerase (PARP), the substrate of caspase-3, is a key marker of apoptosis [21,22]. The expression of the cleaved PARP protein was significantly increased in cells at 24 h after treatment with the combination of LMWF and 5-FU compared with control cells and those treated with LMWF or 5-FU (all *p* < 0.05, Figure 3). However, the expression of the cleaved PARP protein was not significantly increased at 6 h after treatment with the combination of LMWF and 5-FU. Therefore, based on these results, LMWF may enhance the effects of 5-FU on apoptosis of the HCT116 cells through the JNK signaling pathway.

### 2.5. Effects of Signaling Pathways Responsible for LMWF and 5-FU on Decreased Viability of Caco-2 Cells

Caco-2 cells are KRAS wild-type and PIK3CA wild-type colon cancer cells [17]. The cellular mesenchymal–epithelial transition (c-MET) signaling pathway has been reported to be associated with carcinogenesis, invasion, and metastasis in various cancers, including CRC [23,24]. Compared with the expression in normal colon mucosa, the overexpression of c-Met mRNA and protein is significantly increased in CRC tumors [23]. Moreover, the overexpression of c-MET has been reported to be associated with the metastasis and progression of CRC [23]. Furthermore, the c-MET signaling pathway also cross-talks with other cell membrane receptors, including vascular endothelial growth factor receptor (VEGFR) and EGFR [23]. A study reported that the inhibition of the c-MET signaling pathway is a potential treatment modality for various human cancers, including CRC [24]. Therefore, in the present study, we analyzed whether the c-MET signaling pathway is involved in the suppression of Caco-2 cell viability caused by treatment with the combination of LMWF and 5-FU. The expression of the c-MET protein was significantly decreased in cells at 6 h after combined treatment with LMWF and 5-FU compared with control cells and those treated with LMWF or 5-FU (all *p* < 0.05; Figure 4 and Figure 5). Moreover, the expressions of KRAS and p-ERK proteins were significantly decreased in cells at 6 and 24 h, respectively, after treatment with the combination of LMWF and 5-FU compared with control cells and those treated with LMWF or 5-FU (*p* < 0.05; Figure 4). Similarly, the expressions of PI3K and p-AKT proteins were significantly decreased in cells at 6 and 24 h, respectively, after treatment with the combination of LMWF and 5-FU compared with control cells and those treated with LMWF or 5-FU (*p* < 0.05; Figure 5). Therefore, based on these findings, LMWF may enhance the suppressive effects of 5-FU on the viability of Caco-2 cells through both c-MET/KRAS/ERK and c-MET/PI3K/AKT signaling pathways.

### 2.6. Effects of LMWF and 5-FU on Migration of Colon Cancer Cells

The migration of colon cancer cells was assessed using a wound healing assay. Cell migration was significantly suppressed after treatment with LMWF or 5-FU in HCT116 and Caco-2 cells (all *p* < 0.05; Figure 6A,B). Furthermore, cell migration was further suppressed after treatment with the combination of LMWF and 5-FU in HCT116 and Caco-2 cells (all *p* < 0.05; Figure 6A,B). Therefore, LMWF may enhance the suppressive effects of 5-FU on tumor cell migration in HCT116 and Caco-2 cells. 

### 2.7. Effects of Signal Pathways Responsible for LMWF and 5-FU on Suppressive Effects of Migration in HCT 116 and Caco-2 Cells

The proteolysis and disruption of ECM are important in the invasion and metastasis of malignant cells. Matrix metalloproteinases (MMPs) are important proteolytic enzymes involved in the proteolysis and disruption of ECM [4,25]. MMP-2 has been reported to be associated with invasion and metastasis in CRC [4,25]. Therefore, in the present study, we explored whether the MMP-2 signaling pathway is involved in the suppression of migration in HCT116 and Caco-2 cells caused by combination treatment with LMWF and 5-FU. The expression of the c-MET protein was significantly decreased in cells at 6 h after combination treatment with LMWF and 5-FU compared with control cells and those treated with LMWF or 5-FU in HCT116 cells (*p* < 0.05; Figure 7) and Caco-2 cells (*p* < 0.05; Figure 8). Moreover, the expression of the MMP-2 protein was significantly decreased in cells at 24 h after combination treatment with LMWF and 5-FU compared with control cells and those treated with LMWF or 5-FU in HCT116 cells (*p* < 0.05; Figure 7) and Caco-2 cells (*p* < 0.05; Figure 8). Consequently, according to these results, LMWF may enhance the effects of 5-FU on the suppression of migration in HCT116 and Caco-2 cells through the c-MET/MMP-2 signaling pathway.

## 3. Discussion

Fucoidan has been reported to exhibit with anticancer activities in various cancer types, including CRC [6,7,8,9,10,11,12]. Moreover, fucoidan was reported to enhance the inhibitory effects of co-treatment chemotherapeutic medicines for gastric cancer, esophageal cancer, breast cancer, CRC, and lung cancer [10]. When administered in combination, LMWF has been demonstrated to enhance the effect of chemotherapy and to prevent cachexia-associated muscle atrophy [26]. 5-FU is a critical component of the systemic chemotherapy regimen, commonly used for different malignant tumors, including CRC [14]. At present, EGFR monoclonal antibodies (e.g., cetuximab and panitumumab) are used in combination with a fluoropyrimidine-based chemotherapy regimen to treat patients with mCRC [27]. However, EGFR monoclonal antibodies are effective only for patients with wild-type *RAS* gene mCRC. Moreover, considering the results of our previous study [15], we hypothesize that LMWF augments the anti-cancer effect of fluoropyrimidine-based chemotherapy. Therefore, we performed an in vitro cell lines study in order to elucidate the role of LMWF in enhancing the anti-cancer effect of fluoropyrimidine-based chemotherapy. The mutation statuses of HCT 116 and Caco-2 cells are *KRAS* mutation (p.G13D) and *KRAS* wild type, respectively [17]; therefore, both HCT 116 and Caco-2 cells were used in the present study.

We had used three different doses (600, 800, and 1000 µg/mL) of low-molecular-weight fucoidan (LMWF) to investigate the suppressive effects in cell viabilities of HCT116 and Caco-2 cells. The better suppressive effects of LMWF in cell viabilities of HCT116 and Caco-2 cells were noted at the dose of 800 µg/mL. Moreover, we had used three different doses (5, 10, and 20 µg/mL) of 5-FU to investigate the suppressive effects in cell viabilities of HCT116 and Caco-2 cells. The better suppressive effects of LMWF in cell viabilities of HCT116 and Caco-2 cells were noted at the dose of 10 µg/mL. Therefore, we used the doses of LMWF (800 µg/mL) and 5-FU (10 µg/mL) in the present study. These results will be presented as Appendix A. In the present study, the viabilities of both HCT 116 and Caco-2 cells were significantly reduced by LMWF and 5-FU. Kim et al. reported that the growth of HT29 colon cancer cells was significantly inhibited after administration with 500 μg/mL fucoidan for 48 h [28]. Han et al. reported that fucoidan suppressed the proliferation of HT29 colon cancer cells [4]. Park et al. also demonstrated that the viability of HCT116 colon cancer cells was gradually reduced by treatment with fucoidan [29], and the growth of Caco-2 colon cancer cells was also inhibited after treatment with fucoidan [30]. Moreover, the viability of both HCT116 and Caco-2 cells was further significantly reduced by the combination therapy of LMWF and 5-FU. Therefore, LMWF may enhance the suppressive effects of 5-FU on the viability of both HCT116 and Caco-2 cells. Fucoidan has been studied as a food supplement or a complementary agent to decrease the toxicity and enhance the efficacy of chemotherapeutic agents [31]. Vishchuk et al. demonstrated that fucoidan from the brown alga Saccharina cichorioides could enhance the growth inhibitory activity of resveratrol in HCT116 colon cancer cells [32]. Furthermore, fucoidan is considered a complementary agent that can enhance the efficacy of chemotherapeutic agents in various cancer cells, including lung cancer, breast cancer, and esophageal cancer [31].

In the present study, we demonstrated that LMWF enhanced the suppressive effects of 5-FU on the viability of HCT116 cells through the induction of cell cycle arrest in the S phase and late apoptosis. Park et al. [29] also revealed that fucoidan induced apoptosis and cell cycle arrest in HCT116 cells; however, Park et al. [29] reported that fucoidan induced cell cycle arrest in the G1 phase in HCT116 cells. Kim et al. [28] reported that fucoidan induced cell cycle arrest in the sub-G1 phase in HT29 cells. In this study, we further investigated the mechanism underlying the LMWF-induced enhancement of the suppressive effects of 5-FU on the apoptosis of HCT116 cells, and we demonstrated the involvement of the JNK signaling pathway in the LMWF-induced enhancement. In leukemia cells, the signaling pathway of ERK1/2 and JNK was reportedly involved in fucoidan-induced apoptosis [33]. HCT116 cell apoptosis caused by fucoidan could also be induced by modulating endoplasmic reticulum stress cascades [34]. 

In Caco-2 cells, LMWF enhanced the suppressive effects of 5-FU on cell viability, but LMWF did not significantly induce cell cycle arrest. Although LMWF significantly induced late apoptosis, LMWF did not significantly enhance the effects of 5-FU on late apoptosis induction. We demonstrated that LMWF enhanced the suppressive effects of 5-FU on the viability of Caco-2 cells through both c-MET/KRAS/ERK and c-MET/PI3K/AKT signaling pathways. HT 29 and Caco-2 cells are PIK3CA wild-type colon cancer cells [17]. Kim et al. [35] reported that fucoidan inhibited the viability of HT-29 cells through the insulin receptor substrate-1 (IRS-1)/PI3K/AKT pathway. Han et al. [4] also revealed the suppressive effects of fucoidan on HT 29 cells through the PI3K-Akt-mTOR signaling pathway. Narayani et al. [30] demonstrated that fucoidan induced the apoptosis of Caco-2 cells by increasing the production of reactive oxygen species.

The c-MET signaling pathway has been reported to be associated with invasion and metastasis in CRC [23,24]. MMP-2 is an invasion-related protein; thus, we detected the expression of MMP-2 in order to investigate the effects of LMWF and 5-FU on the migration of tumor cells. In the present study, we demonstrated that LMWF enhanced the suppressive effects of 5-FU on tumor cell migration in HCT116 and Caco-2 cells through the c-MET/MMP-2 signaling pathway. Han et al. [4] also reported that fucoidan suppressed the migration of HT29 colon cancer cells by reducing the expression of MMP-2. In the mouse hepatocarcinoma cell line Hca-F, the expression of MMP-2 was downregulated by fucoidan [36].

There are several limitations in the present study. First, the cellular lines (HCT116 and Caco-2 cells) used in the present study were only categorized by the *KRAS* gene. Second, we did not perform a screening of drugs toxicity that could also lead to half maximal inhibitory concentration (IC_50_) dose determination.

## 4. Materials and Methods

### 4.1. Preparation of LMF

The LMF used in the present study was extracted from S. hemiphyllum and was prepared by Hi-Q Marine Biotech International Ltd. (Taipei, Taiwan), with Good Manufacturing Practice certification. LMF powder has also received the Symbol of National Quality certification in Taiwan. LMF was obtained by enzyme hydrolysis of the original fucoidan, and it has an average molecular weight of 0.8 KDa (92.1%). LMF has a fucose content of 210.9 ± 3.3 mmol/g and a sulfate content of 38.9% ± 0.4% (*w/w*) [14]. LMF powder was dissolved in double-distilled H_2_O and stirred at 65 °C, filter-sterilized through a 0.45-μm MF-Millipore membrane filter (EMD Millipore, Darmstadt, Germany), and stored at −20 °C. 5-FU was purchased from the Sigma Aldrich (Sigma-Aldrich, St. Louis, MO, USA).

### 4.2. Cell Culture

The human colon cancer cell lines HCT116 and Caco-2 were purchased from the American Type Culture Collection (ATCC) (Manassas, VA, USA). All cell lines were cultured in Dulbecco’s modified Eagle’s medium (Gibco, Grand Island, NY, USA) supplemented with 10% fetal bovine serum (Gibco, Grand Island, NY, USA), penicillin (100 units/mL) (Gibco, Grand Island, NY, USA), and streptomycin (100 µg/mL) (Gibco, Grand Island, NY, USA) in a humid atmosphere containing 5% CO2 at 37 °C.

### 4.3. Cell Viability Assay

Cell viability was determined using a 3-(4,5-dimethylthiazol-2-thiazolyl)-2,5-diphenyl-2H-tetrazolium bromide (MTT) assay (Sigma-Aldrich, St. Louis, MO, USA). Cells (1 × 10^4^ cells/well) were seeded into 96-well culture plates and incubated overnight at 37 °C before being treated with LMWF (800 µg/mL), 5-FU (10 µg/mL), and the combination of LMWF and 5-FU. After 24 h, 20 μL of 5 mg/mL MTT was added to each well and incubated at 37 °C for 2 h. The medium was replaced with 100 μL of dimethyl sulfoxide to dissolve the precipitate. Absorbance was measured at a wavelength of 570 nm by a 96-well microplate reader (BioTeK Instruments, Winooski, VT, USA).

### 4.4. Wound-Healing Migration Assay

Cell migration was assessed using a wound healing assay. Cells were cultured in 12-well plates, and when cells formed a monolayer, a 200-μL pipette tip was used to scratch a wound through the entire center of the well by manual scraping. At 0 and 24 h after the wounding, the migrated cells were counted under a microscope. The images of each wounded area were obtained using a bright field microscope, and wound closure was measured with ImageJ software.

### 4.5. Cell Cycle Analysis

Cells (1 × 10^6^) were treated with LMWF (800 µg/mL), 5-FU (10 µg/mL), and the combination of LMWF and 5-FU for 24 h. Cells were subsequently fixed with 70% ice-cold ethanol at −20 °C for 1 h and resuspended in propidium iodide (PI) (Sigma-Aldrich, St. Louis, MO, USA) in the dark at room temperature for 20 min. Finally, the cell cycle distribution was analyzed using a FACScan cytofluorimeter (Becton Dickinson, Franklin Lakes, NJ, USA) with the CellQuest software (BD Biosciences, San Jose, CA, USA).

### 4.6. Analysis of Cell Apoptosis and Necrosis

Annexin V-FITC (Thermo Fisher Scientific, Waltham, MA, USA) and PI (Sigma-Aldrich) were applied for apoptosis analysis. Cells (1 × 10^6^) were fixed with ice-cold ethanol and were treated with LMWF, 5-FU, and the combination of LMWF and 5-FU for 24 h. Cells were subsequently incubated with Annexin V-FITC and PI in the dark at room temperature. The stained cells were analyzed using the FACScan cytofluorimeter (Becton Dickinson, Franklin Lakes, NJ, USA).

### 4.7. Western Blot Analysis

Cells were harvested and lysed with RIPA buffer (Merck Millipore, Burlington, MA, USA), protease inhibitor cocktail (Sigma-Aldrich), and phosphatase inhibitor cocktail (Sigma-Aldrich). Protein samples (25 μg) were resolved on 10% gel SDS-PAGE, and the separated proteins were transferred to a polyvinylidene difluoride (PVDF) membrane (Merck Millipore, Burlington, MA, USA). After the membrane was blocked with 5% skim milk for 1 h, it was incubated with primary antibodies, such as anti-Met (Cell Signaling Technology, Danvers, MA, USA), anti-p-JNK (Cell Signaling Technology, Danvers, MA, USA), anti-JNK (Cell Signaling Technology, Danvers, MA, USA), anti-PARP (Cell Signaling Technology, Danvers, MA, USA), anti-MMP2 (Cell Signaling Technology, Danvers, MA, USA), anti-pAkt (Cell Signaling Technology), anti-pErk (Cell Signaling Technology, Danvers, MA, USA), anti-Erk (Cell Signaling Technology, Danvers, MA, USA), anti-KAS (Abcam plc, Cambridge, England, UK), anti-PI3K (Abcam plc, Cambridge, England, UK), and anti-GAPDH (Abcam plc, Cambridge, England, UK) overnight at 4 °C. After being washed trice with phosphate-buffered saline (PBS) containing Tween 20 (PBS-Tween 20), the PVDF membrane was incubated with the secondary antibody at room temperature for 1 h. After the membrane was washed with PBS-Tween 20, immunoreactive proteins were detected using a SuperSignal™ West Femto Maximum Sensitivity Substrate (Thermo Fisher Scientific, Waltham, MA, USA).

### 4.8. Statistical Analysis

All experimental data are expressed as the mean ± standard deviation. Significant differences between the two groups were determined using Student’s *t* test. A *p* value less than 0.05 was considered statistically significant.

## 5. Conclusions

We demonstrated that LMWF could enhance the anti-cancer efficacy of 5-FU through its effects on tumor cell viability and migration in both colon cancer cell types of HCT116 (*KRAS*-mutated type) and Caco-2 (*KRAS* wild-type). However, the associated signaling pathways are different. In HCT116 cells, LMWF enhances the suppressive efficacy of 5-FU through the induction of cell cycle arrest and JNK-mediated apoptosis. In Caco-2 cells, LMWF enhances the suppressive efficacy of 5-FU through the c-MET/KRAS/ERK and c-MET/PI3K/AKT signaling pathways. In both HCT116 and Caco-2 cells, LMWF enhances the suppressive effects of 5-FU on tumor cell migration through treatment through the c-MET/MMP-2 signaling pathway. LMWF is a potential complementary agent that enhances the efficacy of the fluoropyrimidine-based chemotherapy regimen in CRC with the wild-type and mutated *KRAS* gene. However, further in vitro studies and clinical trials are necessary in order to validate the results of the present study.

## Figures and Tables

**Figure 1 ijms-22-08041-f001:**
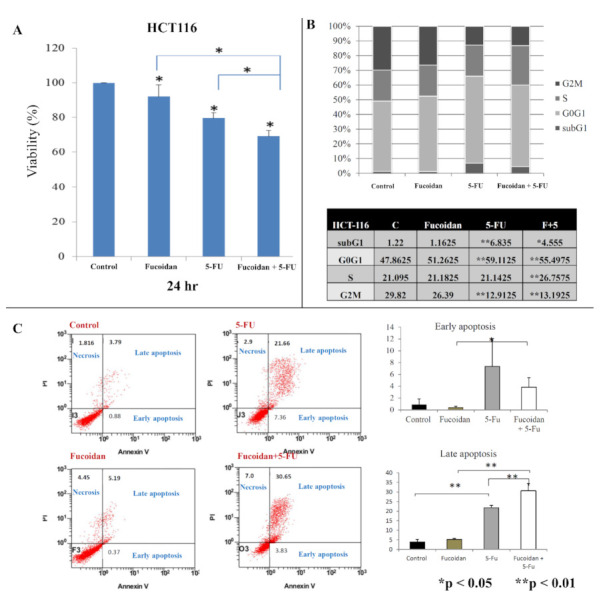
Effects of LMWF and 5-fluorouracil (5-FU) on viability, cell cycle, and apoptosis status of HCT116 cells: (**A**) cell viability of HCT116 cells at 24 h; (**B**) cell cycle of HCT116 cells; (**C**) cell apoptosis of HCT116 cells. Data are presented as the mean of three independent experiments. * *p* < 0.05; ** *p* < 0.01.

**Figure 2 ijms-22-08041-f002:**
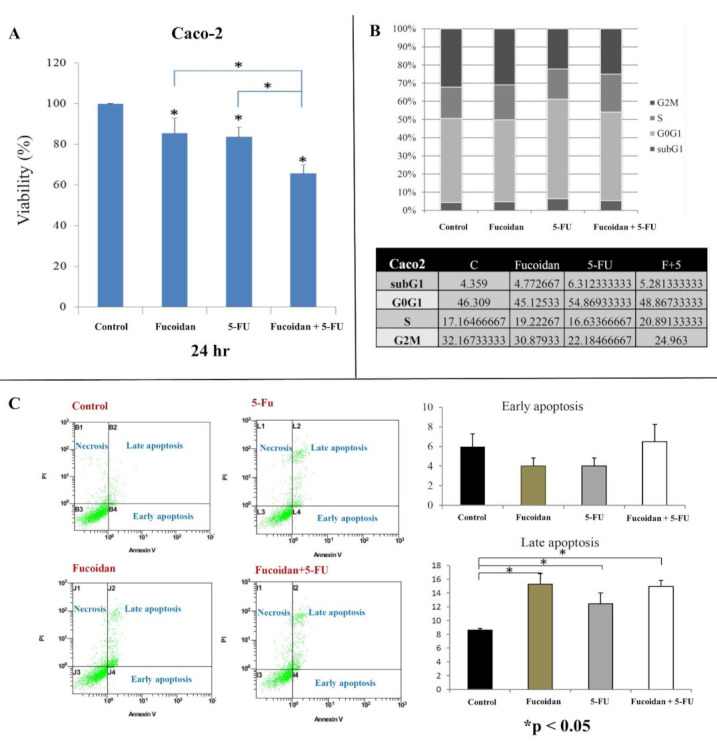
Effects of LMWF and 5-fluorouracil (5-FU) on viability, cell cycle, and apoptosis status of Caco-2 cells.: (**A**) cell viability of Caco-2 cells at 24 h; (**B**) cell cycle of Caco-2 cells; (**C**) cell apoptosis of Caco-2 cells. Data are presented as the mean of three independent experiments. * *p* < 0.05.

**Figure 3 ijms-22-08041-f003:**
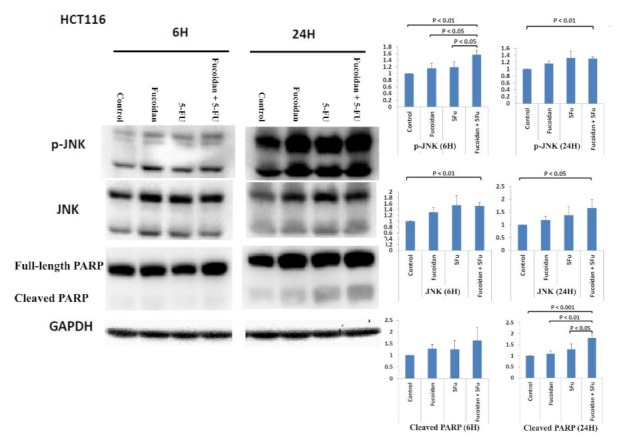
Expression of p-JNK protein, JNK protein, PARP protein, and cleaved PARP protein in HCT116 cells after treatment with LMWF, 5-FU, and the combination of LMEF and 5-FU. Data are presented as the mean of three independent experiments.

**Figure 4 ijms-22-08041-f004:**
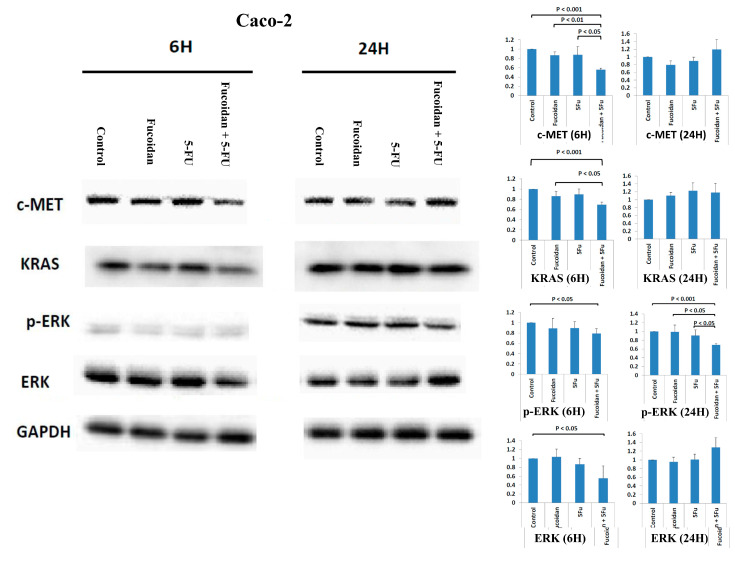
Expressions of c-MET protein, KRAS protein, p-ERK protein, and ERK protein in Caco-2 cells after treatment with LMWF, 5-FU, and combination of LMEF and 5-FU. Data are presented as the means of three independent experiments.

**Figure 5 ijms-22-08041-f005:**
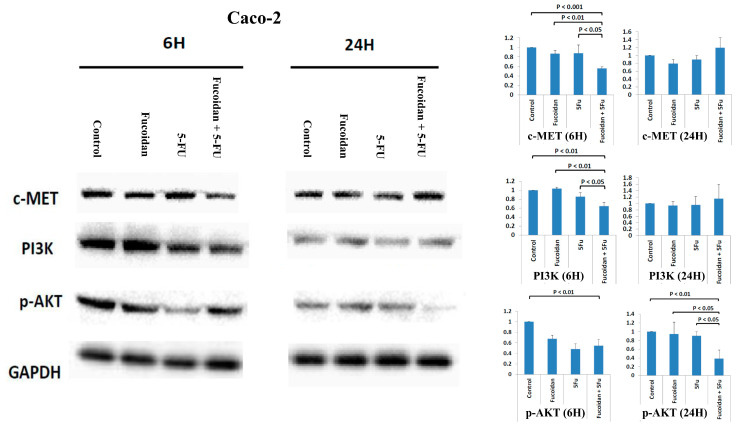
Expressions of c-MET protein, PI3K protein, and p-AKT protein in Caco-2 cells after treatment with LMWF, 5-FU, and combination of LMEF and 5-FU. Data are presented as the means of three independent experiments.

**Figure 6 ijms-22-08041-f006:**
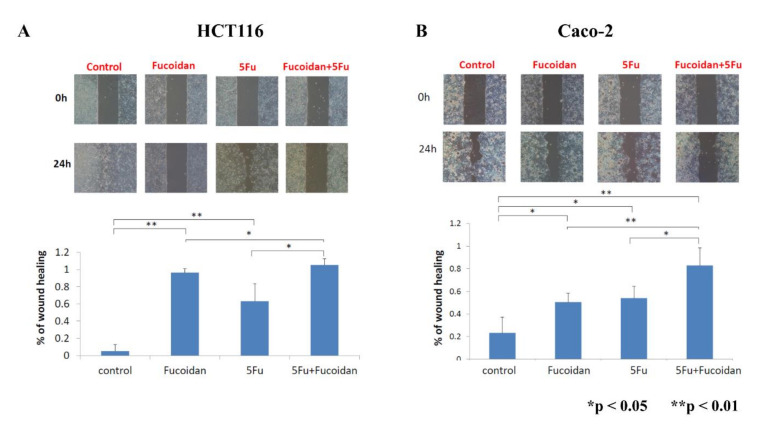
Migration of colon cancer cells after treatment with LMWF, 5-FU, and combination of LMEF and 5-FU: (**A**) HCT116 cells; (**B**) Caco-2 cells. * *p* < 0.05; ** *p* < 0.01.

**Figure 7 ijms-22-08041-f007:**
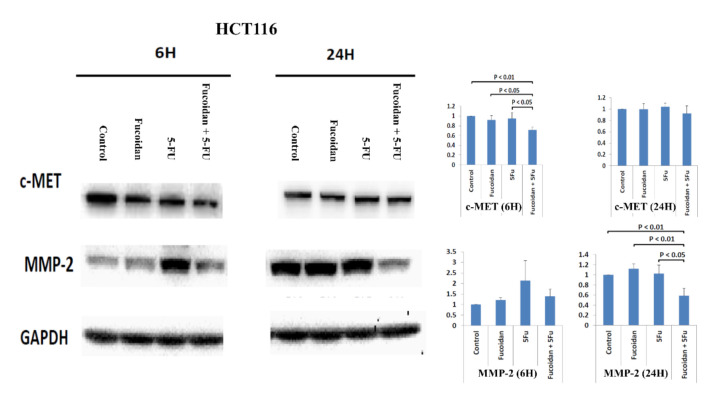
Protein expressions of c-MET and MMP-2 in HCT116 cells after treatment with LMWF, 5-FU, and combination of LMEF and 5-FU.

**Figure 8 ijms-22-08041-f008:**
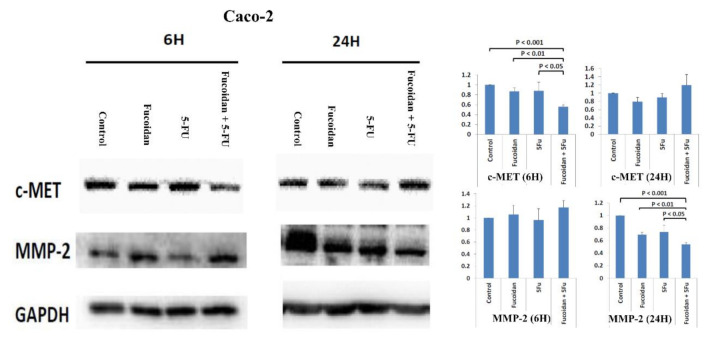
Protein expressions of c-MET and MMP-2 in Caco-2 cells after treatment with LMWF, 5-FU, and combination of LMEF and 5-FU.

## Data Availability

Not applicable.

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
