# Peer review of "Low-Molecular-Weight Fucoidan as Complementary Therapy of Fluoropyrimidine-Based Chemotherapy in Colorectal Cancer"

_ijms, 2021, doi:10.3390/ijms22158041_

Round 1

Reviewer 1 Report

The authors present a paper about  low-molecular-weight fucoidan as a complementary treatment to enhance fluoropyrimidine based chemotherapy in colorectal cancer. The paper is well written and structured. Results support the author conclusions

English in whole manuscript needs revision

The mayor concern about this paper is the use of only cellular line by type regarding KRAS. It would be interesting to evaluate the results in more cellular lines. Moreover in CACO,  LMWF enhanced the suppressive effects of 5-FU but the mechanism of this phenomenon are not clearly described

Author Response

Response to Reviewer 1 Comments

Point 1: English in whole manuscript needs revision

Response 1: Thank you again for your valuable comments on our manuscript. English in whole manuscript has been edited before submission. We can provide the English Editing Certificate.

Point 2: The mayor concern about this paper is the use of only cellular line by type regarding KRAS. It would be interesting to evaluate the results in more cellular lines.

Response 2: Thank you again for your valuable comments on our manuscript. At present, EGFR monoclonal antibodies are important target medicine used in combination with a fluoropyrimidine-based chemotherapy regimen to treat patients with metastatic colorectal cancer. Because RAS gene status is the vital indictor for the treatment with EGFR monoclonal antibodies, we use the cell lines categorized by KRAS gene. It would be interesting to evaluate the results in more cellular lines, and we will state it in the limitations of our manuscript on page 11 Line 293-294.

Point 3: Moreover in CACO, LMWF enhanced the suppressive effects of 5-FU but the mechanism of this phenomenon are not clearly described.

Response 3: Thank you again for your valuable comments on our manuscript. We demonstrated that LMWF enhanced the suppressive effects of 5-FU on the viability of Caco-2 cells through both c-MET/KRAS/ERK and c-MET/PI3K/AKT signaling pathways on Page 10 Line 276-278.

Reviewer 2 Report

The manuscript entitled "Low-Molecular-Weight Fucoidan as Complementary Therapy of Fluoropyrimidine-Based Chemotherapy in Colorectal Cancer" to IJMS journal presents relevant in vitro data that support a synergic effect between 5FU and LMWF treatment, that triggers superior cytotoxic and propapotic effects compared with singular drug-administration.

However, before I thoroughly revise the manuscript, 2 major issues need to be addressed, especially as authors aim to publish their results in a high-quality journal such as IJMS:

First, the manuscript lacks argumentation on how the in vitro working dose selection was performed for cells treatment. Why a single concentration/drug was assessed and the authors did not use multiple doses? Why authors did not perform a screening of drugs toxicity that could also lead to IC50 dose determination?

On the other, the results graphical representation in figures 1,2,3, and 5 are extremely poor in terms of quality, an issue that did not allow me to correlate the graphs with the interpretation of the results and evaluate the accuracy of results interpretation. Therefore, I recommend to i) improve figures quality and ii) break the figures into multiple figures to avoid crowding (consider other ways for data better representation or moving results to supplementary materials.

Round 2

Reviewer 1 Report

All comments were addressed by authors

Reviewer 2 Report

The revised version addressed all my concerns. Therefore, I would recommend the publication of the article in the updated form. Congratulations to the authors!